# A Pragmatic Feasibility Trial Examining the Effect of Job Embedded Professional Development on Teachers’ Capacity to Provide Physical Literacy Enriched Physical Education in Elementary Schools

**DOI:** 10.3390/ijerph17124386

**Published:** 2020-06-18

**Authors:** Chris Wright, John Buxcey, Sandy Gibbons, John Cairney, Michelle Barrette, Patti-Jean Naylor

**Affiliations:** 1School of Exercise Science, Physical and Health Education, Faculty of Education, University of Victoria, Victoria, BC V8W 3P1, Canada; jbuxcey@uvic.ca (J.B.); sgibbons@uvic.ca (S.G.); myb@uvic.ca (M.B.); pjnaylor@uvic.ca (P.-J.N.); 2School of Human Movement and Nutrition Sciences, University of Queensland, Brisbane, 4072 Queensland, Australia; j.cairney@uq.edu.au

**Keywords:** physical literacy, physical education, professional development, physical activity, in-service teacher training, children, elementary school, teachers

## Abstract

A gap in physical literacy (PL) oriented professional development (PD) for generalist teachers exists and thus their capacity to develop PL and maximize student health is potentially limited. We explored the feasibility of a novel job-embedded professional development (JEPD) program (10 weeks) and its impact on teachers’ capacity to deliver PL-enriched physical education (PE) and student PL. A pragmatic feasibility trial with mixed methods included quantitative measurements of teacher PL, knowledge and confidence (pre), and knowledge, confidence, satisfaction and intention (post), as well as self-reported change, to evaluate the impact on teacher capacity and practices. A pre–post comparison of student PL outcomes (motor skills using PLAYbasic, Sport for Life, Victoria, BC, Canada) during the JEPD and teacher implementation phase explored the impact on student PL. In total, 15/44 teachers participated in surveys and 11/44 completed interviews (87% female, mean age bracket = 25–44 years). Confidence to deliver PL enhancing PE increased significantly after JEPD (*p* < 0.0001). Teachers were highly satisfied with the JEPD (*X* = 4.67/5) and intended to change their practices (*X* = 4.09/5). At three months, teachers reported changes including enhanced lesson planning, increased activity variety (often from the JEPD), intentional skill development, student-focused discussions, introductory, transition, and closing activities, and more equipment adaptations. During JEPD, with the exception of throwing (*p* < 0.0001), children’s (47% female, mean age = 7.9 (1.7)) change in running, jumping, kicking and balance walking backwards did not differ from usual practice (UP). During teacher implementation, motor skill competence regressed; confounding factors could not be ruled out. JEPD appears feasible and effective for changing teacher capacity to deliver PL and enhancing PE; however, post-JEPD teacher implementation and outcomes need further exploration.

## 1. Introduction

The school setting is recognized as a key environment to provide opportunities for physical activity for all children, regardless of their socioeconomic status, culture or community [1]. As such, quality physical education (QPE) in school is important to all children’s acquisition of movement confidence, appropriate physical competencies for their developmental age and the knowledge and attitudes to be motivated to be active for life [2]. In conjunction with this, the member states of UNESCO unanimously supported the enactment of the Kazan Action Plan, which states that “fostering QPE and active schools needs provision that is varied, frequent, challenging, meaningful and inclusive” [3]. These are all key components of physical literacy (PL) and attributes that contribute to being an active citizen [3]. PL is defined as “the motivation, confidence, physical competence, knowledge and understanding to value and take responsibility for engagement in physical activities for life” [4]. However, numerous studies have noted a gap in available PD support for PE, and therefore the generalist educators that teach the subject may lack the confidence and knowledge to effectively teach PE [5,6,7,8,9] and consequently their capacity to develop the PL of their students.

Elementary school classroom teachers recognize that a lack of knowledge stems from an absence of specialist training for PE, and uncertainty about what to do and how to do it [7]. Adequate preparation and resources related to QPE facilitation are consistently highlighted as an issue, and this gap in preparation is reflected in a reported lack of continuing PD opportunities for qualified teachers related to PE [5,9,10]. Invoking long-term practice change in teachers extends beyond pre-service education and thus highlights a clear need for continuing PD for in-service teachers to promote ongoing learning and improve teaching practice [11]. Although it is thought that PE specialists provide more effective PE programs, evidence suggests that generalist teachers, with the right support, can also teach effective PE that provides rich movement experiences that develop PL [9,12]. If the outcome of PE is to develop and foster PL within students, it is crucial that effective PD focusing on the development of PL also occurs inside the context of a classroom setting in order to effectively help teachers operationalize PL concepts [13,14,15,16]. PD can be used to help teachers understand how to use pedagogical models to create an environment within their class that fosters PL. This can be done through various means as follows: through using games and activities to create deliberate tasks that challenge and engage students at their own level to increase success and therefore confidence and motivation; spending sufficient time in intentional practice to acquire the physical competency being developed; maximising the number of practice opportunities by using small side tasks and limiting competition to increase competency development in a wide range of contexts and situations; planning for task extensions/refinements and modifying space and/or equipment to aid in individual success and therefore increasing confidence and motivation; and finally by providing assessments that track student learning and progress in order for the children to understand and develop a knowledge of the benefits of physical activity and PE [17]. Effective PD consists of characteristics such as being supportive, job-embedded, instructionally focused, collaborative and ongoing in order to create an experience that is relevant and authentic for each teacher [18]. Effective PD supports teachers through considering not only their needs, concerns and interests, but that of the school and the school district too [18].

PD becomes relevant to teachers when it connects to the responsibilities and practices of everyday activities [18]. As such, situating PD within the school day enables teachers to consider the possibilities of the implementation of what they have learned, try new things and analyze the effectiveness of their actions [18]. JEPD has become a preferred method for knowledge translation in the educational setting [19] because teachers can observe enhanced student outcomes, which is what the vast majority say is the motivation for becoming a better teacher [14]. A quality JEPD program allows for the sharing of what teachers know, what they want to learn and for teachers to connect new concepts, strategies and knowledge to their own unique context and classrooms [20]. The delivery of JEPD is important to learners being able to solve problems in context, and for providing opportunities for feedback and discussion on performance, which enhances the potential for implementation fidelity [21]. The nature of JEPD aligns with the consensus on quality PD characteristics such as content focus, active learning, coherence, adequate duration and collective participation, all elements which are critical to increasing teacher knowledge and skills and improving practice [22]. Thus JEPD constitutes a powerful potential lever to advance student learning and enhance teacher professional knowledge and skills [19].

JEPD research to date has examined its effect on teachers in non-PE subjects [23,24,25,26]. Little PL research has focused on PD with an embedded approach, or it has predominantly focused on the acquisition of movement skills [5,16,27,28]. To our knowledge, research has not yet integrated the contextual learning of teachers from PD delivered by experts in PL embedded within a PE class and tested their impact. The primary aim of this study was to explore the feasibility of a novel PL-focused JEPD program and its impact on teachers’ capacity to deliver and implement PL-enriched PE. A secondary aim was to conduct a preliminary exploration of the impact of self-reported changes in capacity and implementation on student PL (primarily motor skill competence).

## 2. Materials and Methods

### 2.1. JEPD Intervention

Teachers participated in 5 h of JEPD over 10 sessions (1 term) of their PE class. Although some research has indicated that 5 h is an appropriate length [29], the length of JEPD was selected based on pragmatic constraints of the implementation setting including: available resources such as funding, facilitator availability, length of term and teacher PE class schedule. As such, 5 h was the maximum amount of time available for each teacher in the school over a school year. The JEPD was delivered during class time, approximately 30 min for 10 consecutive weeks, by local experts in PL-enriched program delivery from the Pacific Institute for Sport Excellence (PISE). PISE has a reputation as one of the leading facilitators of activities that develop PL within British Columbia. The content of the sessions consisted of games and activities that developed competence in movement skills and built confidence, motivation and knowledge of physical activity in the children. Children’s motivation and enjoyment has been shown to be higher when skills are learned in a games-based context, where the focus is not explicitly on the skill development itself [30]. Activities and skills covered included teaching cues for running, jumping, throwing and catching, as well as other movements such as galloping, hopping, striking and dribbling. The activities were based on various pedagogical models for QPE [31,32] and were used in order to demonstrate the rationale, theory and technique development in a practical way to the teachers. See Table 1 for the strategies and rationale used throughout each session and Table 2 for a sample session. Two PISE staff members facilitated the JEPD and had different roles in the group, with one allocated to facilitate the games and activities and the other to engage the teacher in observation and discussion relating to the session. Different themes were used throughout the program to provide teachers with activities that could be easily replicated, such as sessions where no equipment was used, or activities that could easily be translated to other areas of the school. Each teacher experienced planned games and activities that were developmentally appropriate for their grade level. For example, the same activity may have been played, such as a balance game, but instruction, expectations, difficulty levels and task outcomes were all adapted to be age appropriate. Additionally, within the JEPD, developmental level and ability levels were taken into account and adjusted for in class as necessary to create a student-centred, authentic and diverse experience for each individual. For example, equipment modifications and game adaptations were demonstrated during the class relative to the children in that class. Moreover, the expert facilitators employed different teaching styles throughout the sessions, such as “practice” and “reciprocal” to develop the different domains within PL [33]. Teachers were also encouraged, but not required, to engage in the activities and games they were observing in order to gain a greater understanding of the activity. In addition, all teachers were provided with online resources to supplement the in-class activities and provide more information on PL and a QPE environment, including lesson plans and external documents from organizations such as PHE Canada and the Canadian Paralympic Committee, which all included activities and information for multiple grade levels.

### 2.2. Design

A pragmatic, feasibility trial with mixed methods and a quasi-experimental design including baseline and follow-up questionnaires and post JEPD interviews were used to address the primary aim. The primary outcomes were changes in confidence of the teachers after the JEPD as well as reported teacher knowledge, satisfaction, intention to change their teaching practice and self-reported implementation at 3 months. Qualitative interviews also explored facilitators and barriers to implementation. To examine the second aim, a quasi-experimental pre-post comparison trial was used. This assessed the impact of the JEPD on one component of children’s PL: motor skill competence, compared to usual practice (UP) PE delivery (Fall Phase) and then whether any gains in motor skill competence were maintained or enhanced during a post JEPD teacher implementation phase. Assignment of classes to Fall JEPD intervention (experimental group) or Winter JEPD intervention (UP wait-list comparison) was performed by an administrator at each school based on scheduling and workload. The pre-test measurements occurred on week one, and post-test measurements took place on week ten between September and December (2018). Teacher implementation phase measures were completed with the experimental group after the 10 week period where the UP wait-list comparison group received the intervention in March 2019. The replication of the intervention allowed for comparison of outcomes across phases. Figure 1 shows the process of recruitment, consent, intervention, and analysis for both teacher and student participants. Table 3 introduces the timeline of the study. The study was approved by the University of Victoria Human Research Ethics Board under protocol # 17-110.

### 2.3. Participant Recruitment

All teachers in schools that were participating in the JEPD were asked to participate in the study. Out of 44 teachers, 23 consented to be involved in the data collection and returned signed consent forms (52%), with 15 returning both surveys (34%) and 12 participating in interviews (27%). Regardless of their involvement in the research, all teachers received the JEPD. For the child participants, consent forms were sent home to each child’s parent or caregiver and verbal assent was obtained from the children prior to data collection. If the child did not return a signed consent form, or verbally agree to participate on the day of data collection, they were free to continue in the class without participating in the data collection process. Out of a possible 911 children, 631 returned signed consent forms (69%). The children’s mean age was 7.8 years (range = 4.7–11.0).

### 2.4. Data Collection

Teachers (who were not blind to their allocation status; either Fall or Winter JEPD) were provided a questionnaire before the first JEPD date and asked to return it to the school administrator’s office prior to the first JEPD session where a member of the research team picked them up. After the intervention, this process was repeated. The pre-questionnaire consisted of 10 questions, with questions about knowledge and confidence each having their own subset of questions relating to more specific areas of physical literacy. Each question was answered on a 5-point Likert-type scale, from “no confidence/knowledge” to “a lot of confidence/knowledge.” Post-questionnaires contained similar confidence questions, as well as additional questions regarding teachers’ post-JEPD confidence to apply what was learned and to promote PL-specific concepts with students, their intention being to integrate PL into their practice and their satisfaction with the JEPD experience. We adapted items from other physical activity tools [34,35], and those implemented in past training initiatives to measure physical literacy knowledge and confidence with a demonstrated ability to detect changes [36,37]. The post-workshop survey measured perceived access to resources and intention strength, a construct based on previous work by Rhodes and colleagues [34,35]. Overall intention was measured by three items (intention strength, perceived behavioral control and motivation) using 5-point Likert scales and anchored to a three month time frame. Internal consistency for the questions was established by Hassani et al. [36] and validity for a single intention item by Rhodes et al. [34] Due to the pragmatic nature of the project and related timelines, we could not establish the reliability and validity of the “adapted” tools or confidence measures.

Post-program interviews (*n* = 12) were arranged and data were collected in-person by a member of the research team using a semi-structured interview guide that provided structure across participants while allowing for unanticipated responses. Questions investigated their experience in the JEPD intervention specific to this analysis. Extensive hand-written notes captured the exact terminology, colloquialisms, and labels used by the teachers. The notes were shared with the interviewee to check for completeness and accuracy.

Child level motor skill competence was assessed during the first and last class of the intervention in the gymnasium where the class was taking place. This allowed for participants to take part in the data collection and then return to their PE lesson. Two trained individuals assessed the children as they performed the tasks outlined in the PLAYbasic tool [38], namely: run there and back, hop, overhand throw, kick ball and balance, walk backwards. The raters, both with previous experience in motor skill analysis, completed 3 h of classroom training using standard videos and live demonstrations to practice rating, compare scores and adjust where necessary. Agreement was then assessed qualitatively in a 1 h live in-school testing situation where PLAYbasic scoring was reviewed for discrepancies and discussed. During data collection the trained individuals recording the scores stood in different positions, and completed all pre and post data collection from these positions for all children (see Figure 2). Only one rater was blind to classroom intervention allocation. Scores were recorded using paper versions of the PLAYbasic scoresheet and the mean of the two rater’s scores were analyzed.

### 2.5. Data Analysis

Data were analyzed using SPSS version 24 (International Business Machines Corporation, Armonk, NY, USA). Descriptives and related sample t-tests were used for the teacher data and child data between follow-up and teacher implementation phase. A repeated measures ANOVA was used for child level data to explore time and time by condition effects from baseline to follow-up (Fall phase).

The interview notes were transcribed verbatim into typed pages for the purpose of analysis and data were categorized into codes and categories independently by two members of the research team experienced in qualitative research methods [39]. Conventional inductive content analysis was used, as existing theory and research literature on embedded PD addressing PL in an elementary PE setting were limited [40,41,42]. It provided a systematic method to classify the text into discrete groups [39,41]. Subsequent to independent coding, the researchers met to discuss similarities and nuances amongst the initial codes to solidify the categories. If there was divergence on categories, discussion was initiated to reach agreement. Repeated review and discussion occurred as categories were developed [43,44]. Using inductive analysis and constant comparison ensured the reliability of the coding.

The trustworthiness of the data was established using several methods. First, the handwritten interview notes were reviewed by the teachers in order to strengthen trustworthiness through member verification. Then, each member of the coding team also provided peer debriefing by questioning assumptions and ensuring evidence for decision making was thorough [43]. Triangulation was incorporated three ways: analysis of themes and categorization by two researchers; across participants with various levels of experience; and by utilizing multiple data collection methods (pre and post surveys and face-to-face interviews). Finally, trustworthiness was enhanced by reviewing the data and actively looking for negative evidence [45].

## 3. Results

### 3.1. Teacher Data

#### 3.1.1. Demographics

Of the 15 teachers that participated in the survey, 13 (87%) were female with 67% between 25 and 44 years of age and 33% between 45 and 64 years of age. Their average years of teaching was 14 years (range 2–30 years). Eighty percent (80%) had training on physical activity, 40% on PL, 60% on fundamental movement skills, 20% on sedentary behaviours and 13% on other activities related to PL (globally or other movement skills specifically such as dance). Eighty-seven percent (87%) of the teachers received their preparation through a University course and 13% from ongoing PD. Only two teachers reported no PA preparation. Teachers in the selection taught a variety of grades from Kindergarten through to Grade 5.

#### 3.1.2. Capacity to Deliver PL Enhanced Programming

Teachers’ overall PE teaching skill confidence increased significantly between baseline and follow-up but as illustrated in Table 4, their confidence in being able to adapt activities to different ages, abilities and cultures did not. Teachers’ perception of whether they had the resources they needed to promote PL through their programming was also not significantly different.

At follow-up, teachers’ confidence in their ability to program activities that promoted key components of PL was high (see Table 5). Ninety-three percent (93%) were confident to very confident that they could use what they had learned to improve and sustain PL concepts in their programming. Ninety-three percent (93%) were confident that they could find resources to assist them with PL implementation.

The strength of teacher intentions to integrate PL into their PE programming was high. On average, the teachers’ overall intention to implement PL principles in the next three months was 4.09 (range = 10–15).

#### 3.1.3. Self-Reported Practice Changes

Post-program interviews (*n* = 12/15) showed a variety of changes in practice. Categories of responses illustrated in Table 6 included: more intentional/planned inclusion of games and activities into lesson plans and drawing upon activities they observed during the JEPD, increased variety and skill development, student-focused discussion of, and reflections on, the games and skills, changing introduction and closing games and activities that enhance transitions, and adapting equipment to meet the needs of the children.

#### 3.1.4. Implementation

##### Post Program Satisfaction

Teachers’ overall satisfaction with the JEPD was high; on average 4.67/5 (mode = 5/5; range = 3–5), with 93% satisfied or extremely satisfied. Eighty percent (80%) felt that the JEPD helped them to construct solutions for their practical situation quite a bit to a lot (range 3–5/5).

##### Benefits of JEPD

The most consistently cited benefit of the JEPD across the teachers was the observability. As T4S2 described it “…it is invaluable seeing it in action” while T5S1 said “seeing it in my own space, with my own students so I know exactly what it looks like.”T4S2 emphasized that it was “not just theory”… (it was) “practical” … and “showed how to do it”. T9S1 highlighted that because they were observing “they got to see who could actually skip”. Related further to observability was student enjoyment which was highlighted in a third of the interviews and illustrated by T2S3 who said “kids loved it… were keen…” Several teachers also mentioned qualities of the workshop globally, describing it as “organized really well”—T8S2, “fantastic”—T2S3 and “tailor made for your class…it doesn’t get any better than this…”—T7S3. Specific qualities included the benefits of prolonged engagement and the ability to build lessons over time compared to just having a one-off workshop. For instance, T6S1 emphasized that “PISE built the program gradually—it was doable” while T5S1 said “time… a workshop would never be that amount of time” and T6S1 and T10S1 said they “liked the length (long-term)…” and 10 weeks was “the right amount” respectively. Several teachers mentioned the resources, referring to online access to the lessons and games (google doc links sent via email). Other less-consistently mentioned, but important benefits highlighted the ability to ask questions and discuss with the leader (e.g., “getting to ask questions as game is being played” —T3S2 and the “person was available to chat with”—T1S2). T11S2 highlighted the connection to the PE curriculum, saying that “it touched on all of the curriculum”. Not a benefit but a possible facilitator were qualities of the delivery team. In 9/12 interviews, positive attributes of the leaders were mentioned saying “they were good with kids”—T10S1 and T12S3, “professional knew what they were doing”—T9S1 and “prepared”—T10S1 and T11S2.

##### Challenges with JEPD

Although one half of the interviewees indicated they did not find it challenging to implement what they learned and some indicated they preferred the JEPD method, others highlighted some challenges. Some of these challenges related to the JEPD delivery and included: lack of time to incorporate the theory (“classroom management may detract from theory development”—T3S2), one teacher expressed a preference for more time for discussion with the mentor and another the need for the mentor to be respectfully aware of the existing experience and knowledge of teachers. A further comment addressed leader limitations in their understanding of different age groups of children. Other challenges occurred at the teacher implementation level, regarding the volume of new ideas and activities to integrate (“so much new to me..remembering it all”—T12S3 and “emails were too much”—T10S1), the scheduling of PE activities in an ever-changing school environment (“you have a weekly plan but it seldom goes according to it-something usually comes up”—T9S1), the timing of changes (“changed mid-year rather than at the beginning in September when you plan term”—T9S1), “meeting the needs of children that are developmentally challenged”—T8S2 and “assessing at the individual level while (managing delivery in a large group)”—T3S2.

### 3.2. Child Outcomes

Of the sample of 551 children who were eligible to take part in the study, 283 were allocated to the intervention group and 268 were allocated to the UP PE condition. The total sample consisted of 295 males (53.4%) and 253 females (45.8%) with a mean age of 7.8 years (range = 4.7–11.0). Because of time constraints within the class, 257 children were tested at baseline in the intervention group and 261 in the UP group. Table 7 shows that the distribution of age and sex and overhand throwing skill level did not differ between groups but that run there and back, hop, kick ball and balance walking backwards were significantly higher in the intervention condition compared to controls at baseline.

Table 8 shows that the motor skills of all children (JEPD and UP) improved over time during the Fall phase. However, only overhand throw showed significantly greater improvement in the JEPD intervention group compared to the UP PE condition. Interestingly, once UP children were involved in JEPD in the wait-list comparison phase, the overhand throw improvement was replicated but the improvement in other motor skills was not (see Table 8). During the teacher implementation phase (students in Fall intervention classrooms were re-measured at the same time as the wait-list JEPD intervention), the related samples t-test showed movement competency scores were significantly lower at follow-up (See Table 9).

## 4. Discussion

The primary aim of this pragmatic study was to explore the feasibility of a novel PL-focused JEPD program and its impact on teachers’ capacity to deliver and implement PL-enriched PE. Based on evidence of acceptability, implementation, practicality and changes in a limited set of intermediate short-term outcomes [46], we conclude that it is feasible. Teachers were highly satisfied with the training; they indicated that they preferred the delivery method and most reported few barriers to integrating what they had learned. In terms of practicality, the embedded nature of the intervention reduced known barriers to PD such as availability and time [5,9,10]. The intervention was practical within the resources of one School District, but this may vary by jurisdiction. Finally, the JEPD appeared to have an impact on teacher’s PL-related confidence, their intention to change their practice and subsequent self-reported practice changes. Our secondary aim was to explore student level motor skill outcomes, which showed that significant changes in a manipulative skill during the expert-led JEPD were not sustained during a teacher implementation phase. However, neither the intervention nor the teacher focused solely on motor skills (competence) as physical literacy encompasses a more holistic set of components. The JEPD incorporated QPE classroom management practices and a variety of strategies and activities to address student confidence, motivation and knowledge and understanding. Nor did the research design account for confounders like seasonality. We discuss our findings in the context of the literature and highlight strengths and limitations.

Similar to previous research [6,32], teachers in our study had a “moderate” level of confidence in their PE teaching abilities at the outset and it improved. At follow-up, their confidence in their ability to create PE programming that specifically promoted motor skills, confidence, competence and moderate vigorous physical activity was high. This is important as previous research has highlighted that teachers did not have an accurate understanding of PL, and only 31% of teachers interviewed could clearly articulate the concept [9]. It has also been highlighted that teachers were seeking ways to interpret the curriculum into effective practice, as well as the need for continued PD [9]. The high confidence and intention to change following the JEPD and the improvement in PE teaching skill confidence seen over time supports the use of embedded PD as an effective way to develop teachers’ knowledge and skillset and to interpret the curriculum into practice. Previous research has also shown that given the right supports, generalist teachers are able to deliver a quality PE program [12] and that JEPD provides teachers with knowledge and skills within the context of their classroom, thus highlighting its potential to increase implementation fidelity [19,21]. This research aligns with other studies that have shown that PD implemented outside of the classroom context was effective [16,28,47,48,49].

Interestingly, after 5 h of JEPD intervention, teacher confidence in their ability to adapt physical activities for age, ability and culture was not significantly higher. This may reflect their awareness of the challenges and complexity associated with adapting in a real-world gym class with 20–30 children. In fact, some of the qualitative data highlighted the challenge of incorporating children with diverse needs into games, and this has been highlighted in other studies in Canada and around the world [50,51,52,53]. Conversely, it may also suggest that 5 h was not enough of an intervention dose or that JEPD needed to be enhanced in some way to achieve this outcome (e.g., in situ mentoring).

The child level data showed that the expert facilitation within the JEPD intervention phase had a significant effect on one of the manipulative skills measured. This was seen in both the quasi-experimental comparison phase and replication period when the waitlist teachers participated in the intervention. The positive impact of interventions led by skilled facilitators on fundamental movement skills is consistent with previous research [27,49,54,55]. It should be noted that in the Fall, most of the motor skills improved independent of condition and without true randomization we could not control for outside-of-school physical activity (recreational or competitive sport activity). Although a decline in children’s movement skills during the teacher implementation phase are concerning, we did not have a control condition in place that might have highlighted the impact of seasonal participation in physical activity [56,57]. We also know from the qualitative data that some of the practice changes were focused on enhancing engagement and motivation, more intentional planning and smooth and engaging activities to transition children rather than solely on skill development. These attributes and characteristics within a PE class and the intentional instructional design by the teacher are understood to ultimately develop PL within students [6]. Additionally, the limited range of movement skills measured by the PLAYbasic tool may not capture the development of the children beyond those five narrow competencies. More extensive research is needed to evaluate if teachers’ self-reported changes are measurable in practice and to understand how implementation looks over longer time periods.

Beyond the issues already highlighted, the research findings need to be placed in the context of both the strengths and limitations of the study. First and foremost, this was a pragmatic trial where a school district invested in an intervention model that reduced the barriers that are consistently identified as a hindrance to the development of teacher knowledge in PE, namely time and lack of opportunity [5,58]. Conversely, school district, health and recreation stakeholders leveraged health promotion funding to support the initiative. Although teachers were highly satisfied with the delivery model, this may not be feasible in other jurisdictions. Importantly, the intervention and evaluation were designed to reflect the organizational context in which implementation occurs and this may contribute to enhanced scalability if efficacious [59]. The pragmatic nature of the trial, however, also introduced limitations in that the measurement had to fit into the context of day-to-day school operations (brief teacher surveys, 30-min PE classes located in the gym; interviews conducted on-site during prep periods or lunch). Based on the small gap (2 months) between the School District decision to implement the PL-oriented JEPD and baseline measurement and initial intervention, we adopted and adapted questions for the teacher survey with either: established reliability and/or validity from previous research on PA or training [34,36], or a PL focus and demonstrated sensitivity to changes in knowledge and confidence [37]. We did not establish validity and reliability and this is important for any future efficacy trial and further research on PL related to PD. In terms of the children’s data, we used PLAYbasic as a short form motor skills assessment because more time consuming, comprehensive, previously validated assessment instruments like PLAYfun [60,61], Test of Gross Motor Development-II [62] or Canadian Assessment of Physical Literacy [63,64] were not possible. We did use two raters to enhance validity as per Stearns et al. (2018) and the five skills are part of the validated 12 skill PLAYfun tool. We could not control or adjust for confounders and our findings may not be free from sampling bias based on how groups were assigned. Finally, the use of mixed methods was a strength, allowing us to comprehensively explore the feasibility of the JEPD using quasi-experimental, qualitative and replication approaches.

## 5. Conclusions

A novel PL-oriented JEPD was highly acceptable to and preferred by teachers with few implementation barriers and many benefits. At the teacher and school district level, it was practical and in a limited test of its efficacy, it enhanced generalist teachers’ PL-related confidence, knowledge and intention/motivation to integrate PL concepts. Thus a real world PL-oriented JEPD was feasible, allowing generalist teachers to deliver a PL-enriched PE program. The next steps in the research are to establish the reliability and validity of the PL-oriented JEPD outcome measures and progress to a full randomized controlled efficacy trial. If possible, the efficacy trial should also explore teacher implementation of PL concepts and changes in practice after JEPD and its impact on children more thoroughly. Practical implications emerging from the study include the importance of incorporating JEPD into efforts to integrate PL into professional practice and the ongoing importance of teaching supports (e.g., practical resources, intentional planning tools) and focusing on class management skills for active spaces. With possible school and district level pragmatic limitations on PD time and resources, we recommend an additional focus on embedding PL concepts in the pre-service/teacher preparation learning environment.

## Figures and Tables

**Figure 1 ijerph-17-04386-f001:**
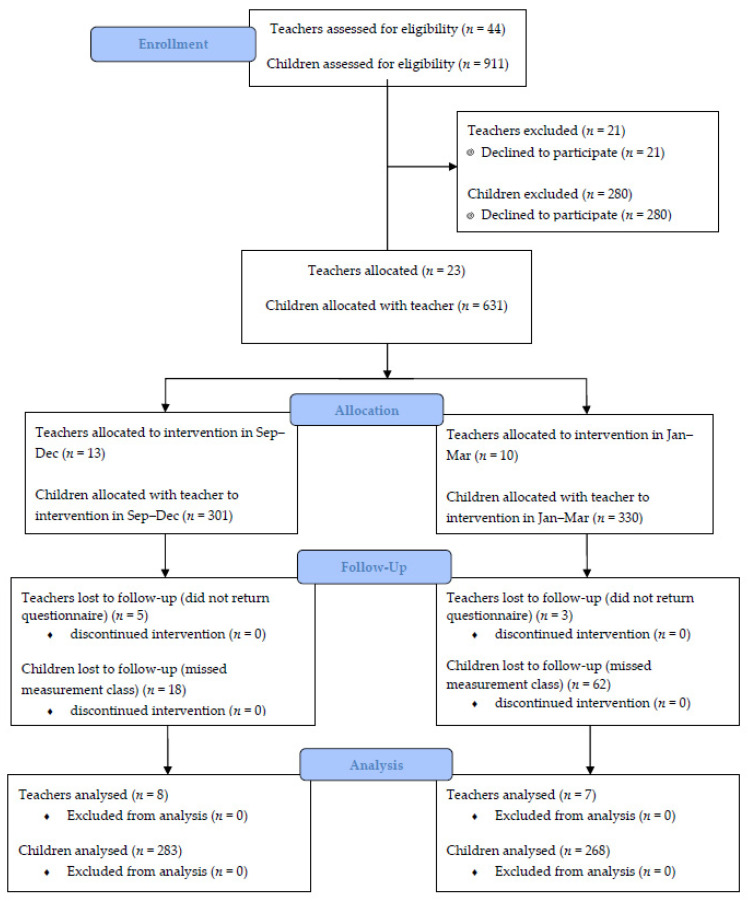
Consolidated Standards of Reporting Trials (CONSORT) table for process of teacher and child level recruitment, consent and analysis.

**Figure 2 ijerph-17-04386-f002:**
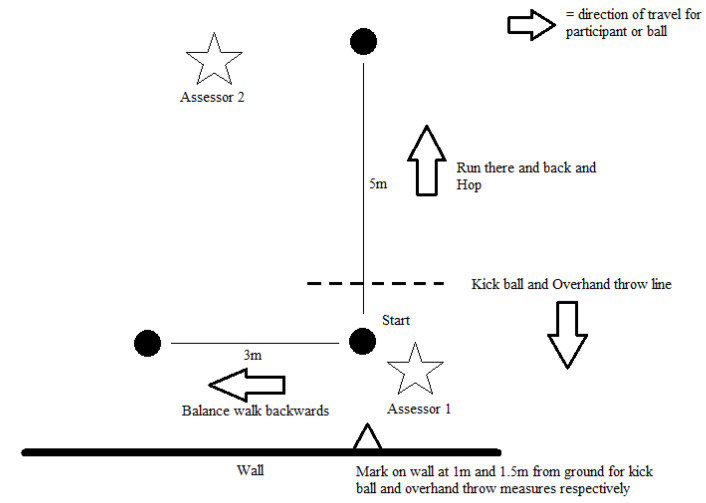
Layout of PLAYbasic assessment protocol for child level data collection.

**Table 1 ijerph-17-04386-t001:** Strategies implemented in each session, their connection to physical literacy and the rationale provided to teachers for their use within a physical education setting.

Strategies Highlighted in Each Session	Connection To Physical Literacy	Professional Development Rationale/Purpose
Transitions	Class management	Engaging children immediately during initial phases of lesson and maintaining engagement between activities
Expectations of session	Class management	Creating a clear understanding and expectation of behaviours within environment
Explain all games	Quality Physical Education	Regardless of previous experience, explaining every activity, every time; creating clear understanding and expectations between all participants
Warm up	Motivation	Moving early to provide purpose for session
Skill development	Physical competence	Progressing skills using games and activities
Practice	Confidence	Allowing time without peer, audience or time pressure
Progressions	Motivation	Increasing or decreasing the difficulty of each activity
Challenge	Confidence	Increasing learning through appropriate amount of success and failure
Modifications	Confidence	Adjusting the activity to engage all students
Individualized	Motivation	Skill development, progressions and modifications are different for all individuals
Small-sided games	Confidence	Allowing for greater interaction with object and other players within a game-play context
Cooperative activities	Confidence	Providing an opportunity to develop problem solving, decision making and communication skills
Spatial awareness	Knowledge and understanding	Enhancing self-awareness and how to orient body within environment and context
Strategy	Knowledge and understanding	Developing decision making skills at speed in contextual practice
Choice	Motivation	Increasing motivation through autonomous decision making and ability to choose level of challenge
Variety	Motivation	Engaging students of varying experiences, ability levels and interests
Peer to peer feedback	Confidence	Allowing students to understand what movement to make and how the movement is formed and can be improved.

**Table 2 ijerph-17-04386-t002:** Example of a Grade 2 throwing lesson, including components related to why each activity was chosen in relation to the knowledge development of the teacher and the connections to PL.

Example of Grade 2 Throwing Lesson			
When	What	How	Why
On entry	Energizer	Touch all 4 walls 5 jumps (style of your choosing) at the circle Balance in any position on one foot and wait for everyone to be ready	Allows students to be active and engaged as soon as they enter the gymnasium while the rest of the class arrives (motivation). Sets the expectation right away that this is a space for activity. The Energizer is consistent each week to allow children to engage without having to wait. Choice is embedded through choosing a style of jump (e.g., knee tucks, jumping jacks, burpees etc.) (motivation).
	Expectations	Ask students if they remember the class expectations: Be respectful Be safe Have fun!	Creates knowledge and understanding of expectations for the environment. Sets out learning goals and behaviours.
Warm up game	Tent tag	Remind students of running cues from previous week Chip from hip to lip Elbow the bad guy behind you If tagged, you must form a “tent” with your body (high plank). To be ‘free’ another person must crawl under your tent. Modifications: More difficult: Only 3 points of contact with the floor Easier: participants can be on their knees until someone tries to save them	Reminders of previous lesson to enhance knowledge. A fun, active game to create engagement in the lesson (motivation). Modifications are presented to create student-centred activities should any individual be unable to perform initial task or want to challenge themselves (confidence and motivation). Cooperation is required in order to untag participants, as well as to trust the tent will stay standing while a person goes through (motivation).
Skill development	Popcorn shooter	Demonstrate and explain throwing cues Make a star with your body Point at your target Throwing arm all the way behind your ear Step with front foot and throw! Participants work as a team to get all of the beanbags or balls into a bucket in the middle of the space. Poly-spots are spread out at varying distances around the bucket which participants must stand on before throwing. Modifications: More difficult: change distance of poly-spots, balance on one foot, use opposite hand Easier: move polyspots closer, larger target to throw into, have multiple targets around the space	Demonstrations and explanations of key components of throwing skills ensure all learners can see, hear and practice required movements (knowledge and confidence through modeling). Game is simple and involves many repetitions of the movement in order to complete (competence). Target should be large enough so all participants can have success (confidence). Modifications are presented to provide choice for individuals with more or less experience with task (confidence), providing autonomy within activity to challenge the individual (motivation).
Practice	Skittles	Have brief discussion with children around what made throwing easier/more difficult. Reinforce elements that made things easier. Two teams are on opposite sides. The blocks are set up in a line on either side. Cones are used to mark out a defending zone in front of the blocks. The object of the game is to knock down the other team’s blocks before all of yours get knocked down. Modifications: More difficult: move skittles further away, use opposite hand Easier: move skittles closer.	Reinforces knowledge. Situates the skill in a more game-based context. Lots of balls and many targets create a small-sided activity where many people are throwing many objects at many targets (as opposed to many people trying to get one object at one target). Modifications increase or decrease challenge in order to create success (competence and confidence). Game-based context enables decision-making and strategy to be developed.
Before transition to class	Calm Bodies	Have all participants find a quiet space on the gym floor, be sure to have their own space (away from walls and other participants). Once their body is calm one of the leaders taps them on the foot, signaling they should quietly line up at the door.	Beginning the transition back to a more sedentary environment. Body and physiology awareness is developed by controlling breathing, while quiet reflection on activities provides opportunity to think about skills and games played (knowledge).

**Table 3 ijerph-17-04386-t003:** Timeline of study from assignment of classes to data analysis.

Date	Objective
June 2018	Classes assigned to Fall or Winter intervention
September 2018	Baseline testing for all children and pre-intervention survey given to all teachers Beginning of intervention
December 2018	Follow-up testing on intervention group Post-intervention survey given to teachers finishing JEPD
January 2019	Follow-up testing on usual practice prior to receiving intervention Interviews with teachers that completed JEPD in Fall 2018
March 2019	Third and final assessment for all children (Fall and Winter JEPD intervention) Post-intervention survey given to teachers in usual practice after receiving intervention
April 2019	Interviews with teachers that completed JEPD in Winter 2019 Data entry and cleaning
May–August 2019	Data analysis

**Table 4 ijerph-17-04386-t004:** Changes in PL related to teaching skills confidence and perceptions of resource availability between baseline and follow-up after Job Embedded Professional Development.

Variable	*n*	Pre-Test M (SD)	Post-Test M (SD)	Statistic	*p* Value
Perception of availability of resources to support their PL programming	15	3.40 (0.63)	3.87 (0.64)	*t*(14) = −1.71	*p =* 0.110
Confidence in ability to:					
Provide opportunities for exploration and free play	15	3.87 (0.52)	4.40 (0.74)	*t*(14) = −2.78	*p* < 0.05
Adapt PA for different ages, abilities and cultures	15	3.40 (0.49)	3.87 (0.63)	*t*(14) = −1.83	*p* = 0.089
Create an environment that promotes PA engagement	15	3.63 (0.40)	4.28 (0.46)	*t*(14) = −5.87	*p* < 0.01
Total Confidence ^1^	15	10.93 (1.33)	12.80 (1.61)	*t*(14) = −3.84	*p* < 0.01

^1^ A sum of all items.

**Table 5 ijerph-17-04386-t005:** Post JEPD teacher confidence in their ability and intentions to promote key PL concepts.

Variable	Mean	Mode	Range (Min–Max)
Confidence in ability to promote: (score range 1–5)			
Locomotor skills	4.13	4.00	3–5
Manipulative skills	4.20	4.00	1–4
Balance/stability skills	4.33	4.00	2–4
Moderate/Vigorous Physical Activity	4.53	5.00	1–5
Children’s confidence	4.00	4.00	3–4
Children’s motivation	4.13	4.00	2–4
Intention to integrate PL: (score range 1–5)			
I will use PL principles (5 = strongly agree)	4.00	4.00	3–5
It will be difficult to include (5 = strongly disagree)	3.93	4.00	2–5
I am motivated to include PL (5 = strongly agree)	4.33	4.00	4–5
Total Strength of intention (score out of 15 (Standard Deviation)	12.27 (1.62)	N/A *	10–15

* N/A = not applicable.

**Table 6 ijerph-17-04386-t006:** Response categories and illustrative quotes from teacher interview content analysis ^1^.

Response Category	Illustrative Quotes
More intentional/planned inclusion of games and activities	“…started to put 2–3 activities together into a short time period” T4S2 “divided kids into groups more often to use more activities” T6S1 “having more of a lesson format…better plan…more cohesive, things that work together” T5S1 “making sure I follow my schedule” T1S2
Increased variety and skill development	“now more familiar with skill development” T8S2 “I copied some of their activities” T5S1 “Learned a different vocabulary” T6S1 “incorporated the game part…skills (are now) game oriented” T1S2 “resource is helpful…(“referred back… to pdf of lessons and games… to remind me”) T6S1 “less sport based, more skill based” T5S1
Student focused discussions/reflections	“I let children choose” T11S2 “involved children … (using) silent thumbs up, thumbs down system’ (that was modeled)” T6S1 “trying to add more skill specific discussion before playing game” T9S1 “more reflecting on games—trying to do a better job of giving back to the students” T4S2
Enhancing transitions with introduction, closing and transitioning activities	“learned a lot of new warm-up and cool down” T12S3 “the games are great way to round up kids” T10S1 “all elements have a closer…I could see how important it is when I saw it…” T1S2
Adapting equipment to meet needs	“I know how to adapt equipment” T8S2 “using more equipment that I had not considered before” T6S1

^1^ Interviews were identified with this system: T# = teacher (# denotes which teacher) and S# = school (code).

**Table 7 ijerph-17-04386-t007:** Baseline demographics and motor skills differences between conditions.

Demographic Measures	Intervention			Usual Practice			*p* Value
	n	M (SD) or n (%)	Range	n	M (SD) or n (%)	Range	
Age (in years)	277	7.9 (1.7)	4.7–10.8	259	7.6 (1.6)	4.8–11.0	*p =* 0.12
Sex (females)	284	133 (47%)		268	120 (45%)		*p =* 0.89
Motor skill							
Run there and back	257	47.4 (12.4)	15.5–77.5	261	42.1 (10.5)	14.5–65.5	*p* < 0.05
Hop	257	47.6 (12.1)	15.5–77.0	261	42.5 (10.2)	0.0–65.0	*p* < 0.05
Overhand throw	257	48.6 (12.0)	9.5–82.0	261	47.5 (8.5)	23.5–80.5	*p* = 0.19
Kick ball	257	45.7 (12.7)	12.0–71.0	261	43.4 (9.6)	20.5–71.5	*p* < 0.05
Balance walk backwards	257	45.9 (15.1)	14.5–75.0	261	43.0 (13.2)	0.0–68.5	*p* < 0.05

**Table 8 ijerph-17-04386-t008:** Results of the repeated measures ANOVA examining changes in motor skills among children involved in the Fall JEPD compared to those in the Usual Practice wait-list condition.

Motor Skill	Condition	Initial		Post Intervention	Time by Condition	*p* Value	Effect Size	Time Effect
		*n*	M (SD)	M (SD)	F	*p*		*p*

Run there and back	JEPD	233	47.6 (12.4)	55.8 (8.0)	0.243	*p* = 0.622	0.091	*p* < 0.05
	UP	240	42.1 (10.5)	50.8 (6.5)				
Hop	JEPD	233	48.0 (12.1)	54.8 (7.6)	0.509	*p* = 0.476	0.095	*p* < 0.05
	UP	240	42.4 (10.3)	49.9 (6.1)				
Overhand throw	JEPD	233	48.7 (11.5)	55.1 (8.3)	19.083	*p* < 0.05	0.032	*p* < 0.05
	UP	240	47.5 (8.5)	50.6 (6.2)				
Kick ball	JEPD	233	45.8 (12.7)	51.2 (8.6)	0.39	*p* = 0.532	0.022	*p* < 0.05
	UP	240	43.4 (9.6)	48.3 (7.4)				
Balance walk backwards	JEPD	233	46.1 (15.2)	54.1 (12.0)	0.35	*p* = 0.554	0.023	*p* < 0.05
	UP	240	43.0 (13.3)	50.3 (9.9)				

**Table 9 ijerph-17-04386-t009:** Paired samples *t*-test between baseline and follow-up for usual practice that received JEPD and for the teacher implementation phase.

Motor Skill	Condition	Baseline M (SD)	Follow-up M (SD)	*t (df)*	*p* Value (2-Tailed)
Run there and back	UP receives JEPD	50.8 (6.6)	51.1 (5.3)	−0.85 (221)	*p* = 0.395
	Teacher implementation	56.0 (7.9)	51.7 (5.3)	10.41 (203)	*p* < 0.05
Hop	UP receives JEPD	50.0 (6.2)	50.4 (5.2)	−1.00 (221)	*p* = 0.318
	Teacher implementation	54.8 (7.5)	51.1 (4.7)	9.64 (203)	*p* < 0.05
Overhand throw	UP receives JEPD	50.5 (6.2)	51.4 (5.3)	−2.61 (221)	*p* < 0.05
	Teacher implementation	55.2 (8.5)	51.8 (4.7)	7.03 (203)	*p* < 0.05
Kick ball	UP receives JEPD	48.3 (7.5)	47.8 (6.7)	1.70 (221)	*p* = 0.090
	Teacher implementation	51.5 (7.8)	49.5 (5.8)	4.95 (203)	*p* < 0.05
Balance walk backwards	UP receives JEPD	50.6 (10.0)	47.6 (9.5)	6.57 (221)	*p* < 0.05
	Teacher implementation	54.2 (11.8)	49.7 (8.9)	8.00 (203)	*p* < 0.05

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
