# Peer review of "A Pragmatic Feasibility Trial Examining the Effect of Job Embedded Professional Development on Teachers’ Capacity to Provide Physical Literacy Enriched Physical Education in Elementary Schools"

_ijerph, 2020, doi:10.3390/ijerph17124386_

Round 1

Reviewer 1 Report

Dear authors,

Thank you for the work done. Your paper about a job embedded professional development on 3 teachers’ capacity to provide physical literacy enriched physical education has many merits and there is clearly a need for this type of studies. Congratulations for the innovative approach taken. However, I some improvements are suggested. Please find my specific comments below.

Abstract

  1. Identify: the number of female/male teachers, mean age and standard deviation, grade levels they were teaching (e.g. grades 1-4)
  2. Identify: the number of male/female students, mean age and standard deviation

Introduction

  1. The introduction sets very well the scientific background of the study. However, the meaning of ‘PL enriched PE’, or how the authors considered that ‘PL looks like in practice’ – which is going to be related to the proposed JEPD intervention – needs to be further clarified.
  2. If appropriate, please specify the study hypothesis.
  3. Page (p) 2, Line (L46). Add a ‘.’ After [3].
  4. Uniform the use of the extended vs abbreviations of the following terms throughout the manuscript: physical literacy, physical education, JEPD, PD.

Methods

  1. P2, L88. Clarify if: 5 hour of JEPD means 30 minutes of PE lessons per week over 10 consecutive weeks.

  1. A critical point that might be improved by the authors is that how did the intervention look like. How is it presented now, seems too diversified (TGFU, CL, SE models; EDUCATION acronym; Team work, para sport… ) and very difficult to allow replication. This is a critical point that in my view that needs to be improved. Can the authors further develop this by giving examples and justifying how all of these approaches come together and are related to PL and QPE?

  1. P3, L97. The reference [32] does not seem to be appropriate to support that ‘children’s motivation and enjoyment has been shown to be higher when 95 skills are learned in a games-based context’. Consider replacing it.

  1. If exist, important changes to methods after pilot trial commencement (such as eligibility criteria), with reasons should be mentioned (e.g. consider teachers questionnaires; PL measurements of students).

  1. I do recommend the authors to add a figure with the model and timeline of the study (in addition to figure1). As a reader I had to struggle to understand it, and I felt this could help to better illustrate what the study and, for example, what was said in p. 3, L127-134.

  1. Of critical importance is the clarification about the validity and reliability of the measures used in the teacher questionnaires. Were their any previous studies that have used those questions? If not, how were these questionnaires built and validated? Clarification on these issues is needed since several analyses are focused on the teacher questionnaires data.

  1. P.5, L163. Correct the eventual typo.

  1. As for the blinding, please clarify: who was blinded after assignment to interventions (for example, participants – teachers, students; those assessing outcomes) and how?

Results

  1. Please consider to uniform the use of ‘Mean’ vs ‘M’ across tables (e.g. se 1 and 2). Suggest using legends for M and SD, for example.
  2. Identify the grade levels of the teachers taught classes.
  3. P6, L205; confirm if you did mean ‘culture did not’.
  4. As for the illustrative quotes identify the teacher (e.g. teacher 1, school A; teacher 4, school B). This allows the reader to see that the quotes came from several teachers and not from a reduced group of teachers (e.g. table 3 and in text quotes).

Discussion

The discussion is of good quality.

  1. P10, L292: JEPD instead of the extended form.
  2. P10, L300: Use ‘[]’ instead of ‘()’.
  3. P10, L303. Correct the citation form of ‘Stoddart and Humbert (2017)[8]’
  4. P10, L314-315. Confirm if you did not want to say ‘was noteffective’ instead.
  5. Add/Improve: the implications for progression from pilot to future definitive trial, including any proposed amendments.
  6. Add/Improve: the practical implications stemming from this trial for how PL looks like in practice; should or could be considered in future interventions of this type.

Conclusion

  1. Could be further improved. Based on the study findings I suggest to remove the sentence ‘and practices; allowing them to deliver a physical literacy 363 enriched PE program’.

References

  1. Please, the references list need to be double-checked. Not all of them are reported in a consistent manner (e.g. 40. Journal is lacking; 47. Not abbreviated form; check others).

Author Response

Thank you for your considered edits and suggestions. Please review the attached document for all comments and edits to the article.

Reviewer 2 Report

The topic raised by the study seems important and relevant to me.
Using or using the mixed method of research I also consider relevant in this type of study.
However, the study may improve if the type of qualitative analysis that is done is expanded. The categorization process, the triangulation process, the methodological complementarity process.
I therefore ask the authors to make an effort to better give and justify the credibility of the qualitative research that is being done.
And also that they strive to present clearly the conclusions of their study.
That the impact achievements are clearly reflected, the authors point out in the objectives.

Author Response

(The authors gave the same response as above.)

Reviewer 3 Report

 Review:

  1. Introduction
  • P2, L80 – reference 17, “without an embedded approach”. This is not true for reference 17 as the authors adopted Hunzicker’s (2011) checklist for effective professional development, whereby ‘job-embedded’ was a key feature. Thus the ‘gap in the research’ sentence following should be slightly amended to reflect this.
  • I would urge the authors to integrate Hunzicker’s (2011) paper into the introduction: Hunzicker, J. (2011). Effective Professional Development for Teachers: a checklist. Professional Development in Education. 37(2): 177 – 179.

  1. Materials and methods

2.1. JEPD intervention

  • After reading this section, it sounds like the JEPD mainly focused on the physical domain of PL rather than promoting a holistic approach. Could the authors outline how this training is different to other QPE training, specifically, elaborate further on how the other PL elements (confidence, motivation, knowledge and understanding) were “taken into account” (P3, L99).
  • Social cohesion – I wondered if the authors considered the Australian Sports Commissioners definition of PL that incorporates the social aspect? It would seem that this definition aligns more with the ‘social cohesion’ aspect than the IPLA definition in the introduction (P3, L103).
  • I would question whether 5 hours is sufficient to i) get the holistic messages of PL across to teachers, ii) train teachers to develop QPE lessons, and iii) focus not only on the content/resources but also the pedagogy that is required to achieve QPE and the outcome of PL. More rationale is required in the section to convince the reader otherwise.
  • Further, more information on how exactly the JEPD is ‘job- embedded’ is required in this section. It is currently reading like a traditional CPD that has some element of follow-up. Again, I signpost the authors to Hunzicker’s checklist.
  • How was the JEPD intervention modified to tailor the different developmental needs/ages of the participants? E.g. 4-11 age range.

2.2. Design

  • P3, L124 – Why only track the physical domain of PL for children? Surely this sends the wrong messages to teachers if the holistic definition was shared with them on what physical literacy is?
  • How much progress is a child expected to make in 10 weeks?
  • How were children matched from the experimental/ UP comparison groups? Age, maturation, random?

2.3 Participant recruitment

  • What was the age of the children that took part? This information should be made available in this section to set the context for the teacher’s responses in section 3. Results.

2.4 Data collection

  • Was the teacher’s questionnaire validated/ piloted? If so, can you outline the procedures to achieve this in this section? If not, how confident are the authors that the validity of the 10 items on the questionnaire?
  1. Results
  • P7, Table 2 – “use PL principles” and “motivated to include PL” – I firstly fear that these are leading questions. Secondly, PL is an outcome of QPE, and therefore teachers cannot truly ‘include PL’ it in their practice, they should embed principles of QPE with the outcome being PL. This is a poorly worded item.

3.1.4.2 Benefits of JEPD

  • Despite teachers acknowledging that the JEPD was better than “having a one-off workshop” (P8, L243), 5 hours of support is still limited. Please can the authors justify somewhere in the paper why this number was selected and how it was deemed sufficient?
  • P8, L255 – change challenge to challenging.
  • P8, L259 – teachers comment on “age of children”. More information is required upfront in section 2 on children’s age ranges and how the JEPD accounted for this, in order to put this teacher’s finding into context (see earlier comment).
  • The children’s age range comes in the next section, but this is too late.

  1. Discussion
  • P10, L292 – “implemented a novel job-embedded professional development intervention” – from the design and the results, there needs to be more explanation in earlier sections on how the JEPD is job-embedded to make this claim. Again, authors should refer to Hunzicker’s (2011) checklist.
  • P10, L293 – “feasible” what criteria did the authors use to

Author Response

(The authors gave the same response as above.)
